# Cachexia and Sarcopenia in Oligometastatic Non-Small Cell Lung Cancer: Making a Potential Curable Disease Incurable?

**DOI:** 10.3390/cancers16010230

**Published:** 2024-01-04

**Authors:** Valentina Bartolomeo, Mandy Jongbloed, Wouter R. P. H. van de Worp, Ramon Langen, Juliette Degens, Lizza E. L. Hendriks, Dirk K. M. de Ruysscher

**Affiliations:** 1Radiation Oncology, Fondazione IRCCS Policlinico San Matteo, 27100 Pavia, Italy; 2Department of Clinical Surgical, Diagnostic and Pediatric Sciences, Pavia University, 27100 Pavia, Italy; 3Department of Radiation Oncology (Maastro Clinic), GROW—School for Oncology and Reproduction, Maastricht University Medical Center, 6229 ER Maastricht, The Netherlands; dirk.deruysscher@maastro.nl; 4Department of Pulmonary Diseases, GROW—School for Oncology and Reproduction, Maastricht University Medical Center, 6229 ER Maastricht, The Netherlands; m.jongbloed@alumni.maastrichtuniversity.nl (M.J.); lizza.hendriks@mumc.nl (L.E.L.H.); 5Department of Respiratory Medicine, NUTRIM Research Institute of Nutrition and Translational Research in Metabolism, Maastricht University Medical Center, 6229 ER Maastricht, The Netherlands; 6Department of Pulmonology, Zuyderland Medical Center, 6419 PC Heerlen, The Netherlands; j.degens@zuyderland.nl

**Keywords:** NSCLC, synchronous oligometastatic disease, intention to treat, progression-free survival, overall survival, sarcopenia, cachexia, toxicity, psoas muscle index

## Abstract

**Simple Summary:**

Synchronous oligometastatic non-small cell lung cancer (NSCLC) represents an intermediate state of metastatic disease with a limited number of metastases. It has been suggested that adding local radical treatment (LRT) in this setting may improve the survival outcomes, but there are no validated tools to better select those patients who are most likely to benefit from LRT. The presence of cachexia and sarcopenia at diagnosis seems to be linked to poorer outcomes in local or advanced NSCLC, but it is not clear if these factors can be used to guide the treatment decisions in oligometastatic NSCLC and to preclude a possible radical treatment in patients with baseline cachexia or sarcopenia. For this reason, we evaluated the impact of cachexia and sarcopenia on survival outcomes and toxicities in a group of patients with synchronous oligometastatic NSCLC on an intention-to-treat basis. Considering the different definitions of sarcopenia used among different studies, we used the Psoas Muscle Index (PMI) as a surrogate of sarcopenia. Progression-free survival was longer for patients without cachexia and sarcopenia compared to those with cachexia and/or sarcopenia.

**Abstract:**

Among patients with advanced NSCLC, there is a group of patients with synchronous oligometastatic disease (sOMD), defined as a limited number of metastases detected at the time of diagnosis. As cachexia and sarcopenia are linked to poor survival, incorporating this information could assist clinicians in determining whether a radical treatment should be administered. In a retrospective multicenter study, including all patients with adequately staged (FDG-PET, brain imaging) sOMD according to the EORTC definition, we aimed to assess the relationship between cachexia and/or sarcopenia and survival. Of the 439 patients that were identified between 2015 and 2021, 234 met the criteria for inclusion and were included. The median age of the cohort was 67, 52.6% were male, and the median number of metastasis was 1. Forty-six (19.7%) patients had cachexia, thirty-four (14.5%) had sarcopenia and twenty-one (9.0%) had both. With a median follow-up of 49.7 months, median PFS and OS were 8.6 and 17.3 months, respectively. Moreover, a trend toward longer PFS was found in patients without cachexia and sarcopenia compared to those with cachexia and/or sarcopenia. In multivariate analysis, cachexia and sarcopenia were not associated with an inferior survival, irrespective of receiving radical treatment. High CRP was associated with inferior survival and could be a prognostic factor, helping the decision of clinicians in selecting patients who may benefit from the addition of LRT. However, despite the homogeneous definition of oligometastatic disease and the adequate staging, our subgroups were small. Therefore, further studies are needed to better understand our hypothesis and generating findings.

## 1. Introduction

Non-small cell lung carcinoma (NSCLC) is the primary cause of death related to cancer, mainly because patients present with advanced disease due to the late onset of symptoms [1]. Within the spectrum of limited and advanced NSCLC, there is a group of patients (approximately 20–50%) presenting with synchronous oligometastatic disease, defined as a limited number of metastases (usually three to five in a maximum of three organs) detected at the time of first NSCLC diagnosis [2,3]. Oligometastatic disease (OMD) is a unique subgroup of stage IV disease, as some patients with OMD can benefit from adding local radical therapy (LRT, i.e., minimal invasive surgery or stereotactic radiation therapy (SRT)) to systemic treatment, with a possibility of long-term disease control or even cure [4,5,6,7]. The addition of LRT to NSCLC responding to systemic therapy in OMD is advised in clinical guidelines, though this is based on limited evidence [8,9,10]. Moreover, immune checkpoint inhibitors (ICI)- and tyrosine kinase inhibitors (TKI)-based treatments are now the preferred first-line treatment strategies and these drugs could be beneficial to patients with OMD treated with radical intent. Preclinically, ICI acts synergistically with SRT because of the immunomodulatory effects of this combination [11,12]. Indeed, in a single-arm phase II study (*n* = 51) evaluating pembrolizumab after LRT, an improvement in PFS was seen compared with historical data [11,13]. Furthermore, in a phase III trial (*n* = 133, NCT02893332) evaluating the efficacy of first-generation epidermal growth factor receptor (*EGFR*) TKI with or without radiation therapy in synchronous oligometastatic *EGFR*-mutated NSCLC, a positive impact on survival was suggested when radiotherapy was added to the TKI [14]. A limitation of this trial and others is the selection bias due to not fully staged patients and in most trials the enrollment of only patients with favorable prognostic or predictive characteristics (e.g., only those responding to systemic therapy or only those with a single metastasis without nodal involvement) [15].

Cachexia and sarcopenia both correlate independently with a decline in prognosis [16,17,18]. Cachexia is linked to systemic inflammation with patients showing loss of skeletal muscle mass that cannot be reversed by nutritional support. This can lead to functional impairment and poorer performance status (PS), and it can reduce the effect of chemotherapy [18]. It is also proposed that cachexia and its related changes in inflammatory parameters have a negative effect on the efficacy of ICI [19]. Additionally, sarcopenia might be associated with poor survival in patients treated with either first-line ICI or EGFR-TKI [20]. In patients with both limited as well as advanced NSCLC the prevalence of cachexia is high and is associated with poor survival outcomes (Table 1) [16,17,18,21]. Of note, in these studies, different approaches to measuring and defining sarcopenia or cachexia were used. Moreover, data are lacking for synchronous oligometastatic NSCLC, and especially for intention-to-treat cohorts.

If cachexia and sarcopenia are also associated with poor outcomes (survival, toxicity) in the treatment of synchronous OMD (sOMD), the incorporation of this information could help clinicians decide whether to pursue a radical treatment or not. Therefore, we evaluated the relation of cachexia and sarcopenia with survival and toxicity outcomes in a fully staged, intention-to-treat cohort of patients with sOMD.

## 2. Materials and Methods

### 2.1. Patient Inclusion and Study Design

We performed a retrospective study in two Dutch hospitals (Maastricht UMC+ and Zuyderland MC) including all newly diagnosed patients with synchronous oligometastatic NSCLC. All weekly thoracic oncology multidisciplinary meetings (MDTs) were reviewed in the period of January 2015 to December 2021. As all patients newly diagnosed with NSCLC have to be discussed at the MDT, no patients will have been missed. We defined sOMD according to the consensus published by the European Organization for Research and Treatment of Cancer (EORTC) stating that sOMD NSCLC has a maximum number of five metastases involving a maximum of three organs, based on baseline fluodeoxyglucose–positron emission tomography-computed tomography (FDG-PET-CT) and brain imaging (magnetic resonance imaging (MRI) preferred). Additionally, patients had to be 18 years or older at the time of diagnosis. All patients for whom the MDT recommended an oligometastatic approach (regardless of received treatment), had no other malignancy within 5 years of NSCLC diagnosis (except tumor in-situ), had no second primary NSCLC, and did not participate in a clinical trial during first-line treatment were included.

The following baseline characteristics were collected from the digital medical files: age, gender, World Health Organization performance status (WHO-PS), weight at diagnosis, weight loss in the last 6 months before diagnosis, length, smoking status and pack years, data on histology, biochemical data within 30 days before the start of treatment or at the date of the first pathological confirmed NSCLC diagnosis, molecular and programmed death-ligand 1 (PD-L1) testing, TNM stage based on the eighth edition of the American Joint Committee on Cancer (AJCC) including the number and location of metastases, the decision of the MDT whether the patient was classified as oligometastatic NSCLC, whether LRT was recommended, and information on first-line systemic therapy. Response after finishing treatment was established according to Response Evaluation Criteria in Solid Tumors 1.1 (RECIST 1.1) [33]. Further data collected were: whether patients actually received LRT, treatment toxicity as documented in the medical files, graded according to the Common Terminology Criteria for Adverse Events (CTCAE version 5.0), date of disease progression, subsequent treatment strategies, and date of last contact/death. The regular definition of cachexia was used: cachexia was defined as an involuntary loss of more than 5% body weight in the past six months or more than 2% body weight loss in patients with a body mass index (BMI) ≤ 20 kg/m^2^ [34].

Sarcopenia is defined as low muscle strength, mass, and function, which could be caused by cachexia or ageing [35]. Sarcopenia is reversible and influenced by multiple factors; therefore, we have chosen to divide cachexia and sarcopenia into groups to truly distinguish their impact on survival as separate entities. The European Working Group on Sarcopenia in Older People (EWGSOP) stated in their consensus paper that the measure of the musculature at L3 can be used as a surrogate marker to define sarcopenia [36]. It was also mentioned that a muscle mass of two standard deviations below healthy adults can be used as a definition of sarcopenia [37]. The PMI was calculated as the cross-sectional area of the psoas muscle (cm^2^) divided by the height (m^2^) [38]. For this reason, in our work we used the PMI as a surrogate of sarcopenia. Using the diagnostic CT if available and otherwise the low-dose CT, the cross-sectional area of the psoas muscle at the inferior aspect of the third lumbar vertebra (L3) was used for calculating the PMI. Due to the absence of validated cut-off values for PMI at the L3 level, we used the unbiased sex-specific lower 25th percentile of the PMI in our cohort of patients, which was 6.05 mm^2^/m^2^ for men and 4.20 mm^2^/m^2^ for women.

### 2.2. Statistical Analysis

Statistical analysis was performed using IBM SPSS version 28 (Chicago, IL, USA). Baseline characteristics were analyzed using descriptive statistics and the groups were compared using the Chi-square test for categorical variables and the one-way ANOVA for continuous variables. PFS and OS were calculated from the date of pathological diagnosis and estimated using the Kaplan–Meier method. The median follow-up was estimated using the reversed Kaplan–Meier method and patients without event were censored at the last follow-up date. Univariate and multivariate analysis of the one-year PFS and OS was performed using the logistic regression model. A *p* < 0.05 was considered as statistically significant.

## 3. Results

### 3.1. Baseline Characteristics of Patients

In total, 439 patients with sOMD NSCLC were identified. Of these patients, 205 were excluded because of: no intention of radical treatment (*n* = 84), another primary malignancy (*n* = 59), no adequate staging (*n* = 32), clinical trial participation (*n* = 25), and no follow-up because of subsequent treatment in another hospital (*n* = 5). Patient selection is depicted in Figure 1. Reasons for the MDT to not advise radical treatment for these 84 patients were: poor clinical condition (*n* = 55), tumor load being too large for radical treatment (*n* = 20), or the patient’s wish for no further treatment after diagnosis (*n* = 9). Three patients had a WHO-PS of 3 at NSCLC diagnosis due to symptomatic cerebral edema from brain metastases, but their clinical condition rapidly improved after treatment with steroids and therefore they were deemed candidates for LRT by the MDT. Of the 234 patients with adequately staged sOMD and an MDT recommendation to treat with radical intent, 142 (60.7%) patients actually proceeded to LRT after a response to induction systemic therapy. Reasons for not proceeding to LRT were: progressive disease after induction systemic therapy (*n* = 46), deterioration of the clinical condition during induction therapy resulting in no response evaluation (*n* = 43), and complete response to induction therapy (*n* = 3).

We analyzed the influence of cachexia and sarcopenia by dividing the patients into groups stratified by the presence of cachexia and/or sarcopenia (Figure 1). Group A consisted of patients without cachexia and without sarcopenia, group B contained patients with only cachexia, group C had patients with only sarcopenia, and group D consisted of patients with both cachexia and sarcopenia. The baseline clinical characteristics of the 234 included patients stratified by cachexia and sarcopenia are shown in Table 2. Except for the differences in BMI, PMI, and weight loss, statistically significant differences were found between the groups for WHO PS, smoking status, serum albumin level, and type of systemic therapy.

### 3.2. Survival Analysis

The median follow-up was 49.7 months (95% CI, 42.4–57.1) for all included patients. The median PFS was 8.6 months (95% CI, 7.2–9.9) and the median OS was 17.3 months (95% CI, 13.9–20.7) (Figure 2).

The median follow-up in groups A, B, C, and D was 49.7 (95% CI, 39.9–59.6), 47.0 (95% CI, 42.2–51.8), 43.1 (95% CI, 29.1–57.2), and 57.8 months (95% CI, 37.6–77.9), respectively (*p* = 0.92). Disease progression occurred in respectively 95 (71.4%), 37 (80.4%), 24 (70.6%), and 17 (81.0%) patients. Subsequently, 96 (72.2%), 39 (84.8%), 26 (76.5%), and 15 (71.4%) patients died, respectively.

The median PFS in groups A, B, C, and D was not significantly different with 9.7 (95% CI, 8.2–11.1), 6.1 (CI 95%, 3.4–8.7), 11.4 (95% CI, 7.8–15.0), and 8.1 months (95% CI, 4.9–11.2), respectively (*p* = 0.18, hazard ratio (HR) 1.1; 95% CI 1.0–1.3) (Figure 3a).

There was no statistical significance in the median OS between the groups: 19.3 (95% CI, 12.3–26.2), 13.1 (95% CI, 9.0–17.2), 17.9 (95% CI, 9.2–26.6), and 15.9 months (95% CI, 2.7–29.1), respectively (*p* = 0.44, HR 1.1; 95% CI 0.9–1.2) (Figure 3b).

We further analyzed the patients by selecting the patients who eventually proceeded to LRT after response to induction systemic therapy.

In our patient cohort, only 60.5% proceeded to LRT after response to induction systemic therapy. Compared with the 64.7% of patients proceeding to LRT in group A (no cachexia and no sarcopenia), a smaller percentage of patients with cachexia only (56.5%), sarcopenia only (55.9%), and both cachexia and sarcopenia (52.4%) proceeded to LRT, although this difference did not reach statistical significance.

The median PFS in the groups was not significantly different with 12.3 (95% CI, 10.1–14.6), 7.6 (95% CI, 5.5–9.7), 13.6 (95% CI, 10.8–16.3), and 9.8 months (95% CI, 7.3–12.3), respectively (*p* = 0.14, HR 1.2; 95% CI, 1.0–1.4) (Figure 4a).

Although numerically higher for group A, there was no significant difference in median OS between the groups: 34.9 (95% CI, 25.3–44.4), 19.1 (95% CI, 9.3–28.9), 25.4 (95% CI, 22.8–28.0), and 27.9 months (95% CI, 12.7–43.1), respectively (*p* = 0.51, HR 1.1; 95% CI, 0.9–1.3) (Figure 4b).

Finally, we also compared PFS and OS between patients of group A (without cachexia and without sarcopenia) versus patients with cachexia and/or sarcopenia (groups B, C, and D combined). Although there was a trend toward a longer PFS, the median PFS in group A was not significantly different from the other groups with 9.7 months (95% CI, 8.2–11.1) versus 7.5 months (95% CI, 5.9–9.2), respectively (HR 0.7, 0.6–1.0, *p* = 0.05). The median OS was not different between group A and groups B, C, and D combined: 19.3 months (95% CI, 12.3–26.2) versus 15.9 months (95% CI, 11.9–19.9) (HR 0.8, 95% CI, 0.6–1.1, *p* = 0.10), respectively (Figure 5).

A univariate and multivariate logistic regression analysis was performed to further evaluate the association of cachexia and sarcopenia and survival both in the total group of patients and subsequently in the group of patients who received LRT after induction systemic therapy. In the total group, in univariate analysis, older age at diagnosis (≥75 years) and male gender were associated with inferior survival, and squamous histology was associated with better survival. In multivariate analysis, only older age at diagnosis was associated with inferior survival (Table 3).

For PFS, univariate analysis showed that male gender and high CRP were associated with inferior survival. In multivariate analysis, high CRP was associated with inferior PFS (Table 4).

In the group of patients receiving LRT we also evaluated TRAE and the best response to induction systemic therapy (Table 5 and Table 6). High CRP (>5 mg/L) was in both univariate and multivariate analysis associated with decreased PFS (Table 6).

### 3.3. Safety Profile

No statistically significant difference in the number of toxicities per grade was observed between the four groups (Appendix A).

## 4. Discussion

According to clinical guidelines, patients with sOMD should receive systemic therapy followed by LRT to all visible disease sites upon disease response [8,9,10]. However, different baseline factors, such as cachexia and sarcopenia, may influence the outcome of patients with sOMD. Although several studies have evaluated the impact of sarcopenia and cachexia on patients with limited or advanced NSCLC, to the best of our knowledge, studies regarding sOMD are not available. Therefore, we evaluated the effect of sarcopenia and cachexia on the survival outcomes (and toxicities) of patients with adequately staged sOMD, treated on an intention-to-treat basis in this preliminary report. In our analysis, we divided the patients into four different subgroups according to the presence of cachexia and sarcopenia in the different possible combinations. Although the definition of sarcopenia is related to cachexia, we decided to consider these conditions as separate entities in our group because other factors such as ageing could have an impact on the presence of sarcopenia [35]. In fact, sarcopenia can be reversible, differently from cachexia, and we supposed that those conditions may had a different impact on the prognosis of these patients. As even in the best prognostic group only two thirds of patients proceeded to LRT, the need for baseline prognostic factors that can help in selecting patient candidates for LRT still remains. The presence of cachexia and/or sarcopenia per se should not preclude patients from a radical treatment approach. This is also reflected in the fact that although patients with cachexia and/or sarcopenia had a numerically shorter median PFS and OS, both factors were not associated with survival in univariate and multivariate analysis, stressing the finding that other factors such as response to induction therapy are more important. Also, the comparison between patients without cachexia and sarcopenia (group A) and patients with cachexia and/or sarcopenia (group B-C-D) did not show significant results.

The small impact of cachexia on the survival in sOMD could be due to the recent implementation of ICI and TKI in the treatment strategies and the beneficial effect of ICI and TKI on survival [11,13,14]. Current studies on the effect of cachexia on the efficacy of ICI in advanced NSCLC have been inconsistent and a clear correlation between cachexia and the efficacy of ICI has not been established yet [19,23,24].

Also, studies on the impact of sarcopenia on advanced NSCLC patients treated with ICI and TKI have demonstrated no differences in survival outcomes according to the presence or not of sarcopenia [19,39,40,41,42,43].

However, to the best of our knowledge, we are the first to also evaluate possible prognostic factors in a group of patients with sOMD on an intention-to-treat basis. In fact, even if in our univariate and multivariate analyses, cachexia and sarcopenia were not significant prognostic factors for survival; in the multivariate analysis we found that older age at diagnosis (≥75 years) was associated with inferior OS. Finally, the multivariate analysis for PFS showed elevated levels of CRP as a significant factor associated with poorer PFS the high CRP. Although both cachexia and sarcopenia are also linked to elevated levels of CRP and IL-6 [19], these results may more likely be explained considering that elevated levels of CRP, LDH, and IL-6 seem to be associated with poor survival in different studies and with diminishing the effect of ICI due to their ability to modulate antitumor immune responses [44,45].

The analysis of all the patients with intention of LRT compared with only patients who received LRT showed similar median PFS and OS in patients with both cachexia and sarcopenia and in patients without cachexia and sarcopenia. This highlights the discussion on how big the impact of cachexia and sarcopenia is in patients with a good WHO-PS and that this specific group of patients with cachexia and/or sarcopenia should be selected for receiving LRT (Table 2). Otherwise, cachexia seems to be more common in patients with poor PS, as the presence of cachexia impacts different aspects of the daily life of the patients, thus conditioning their PS. In fact, different studies have shown a correlation between cachexia and poorer PS, but most of the studies available on cachexia and/or sarcopenia enrolled patients with a good PS [46,47]. Further studies are needed to understand whether cachexia or sarcopenia have an inferior prognostic impact on patients with poor PS.

Regarding the role of the ICI in our group, patients in the cachexia and sarcopenia group received significantly more ICI as induction systemic therapy and this could be a possible explanation for the relatively good survival in this group of patients despite the potential poor prognostic impact of cachexia and sarcopenia [11,13]. As earlier mentioned, the effects of cachexia and sarcopenia on the efficacy of ICI and TKI have been inconsistent, and more and larger data on homogenous groups are needed. Of note, serum albumin could be a possible prognostic factor, as it is suggested that low serum albumin levels could be an indicator of adverse outcomes in ICI therapy due to their effect on systemic inflammation [48,49]. This could be of importance in patients with cachexia, as low serum albumin levels are associated with a low nutritional status and thus cachexia, and larger studies on the correlation between low serum albumin levels and the efficacy of ICI are warranted. However, as we collected patients from 2015 to 2021 (in part before the introduction of ICIs), in our population only 15% of patients received treatment with ICIs and 10% of patients received treatment with TKI. This may represent a relevant limitation for the application of our study in the current clinical scenario. In fact, for patients who are candidates to receive ICIs, immunotherapy represents a safe and with long-term tumor-control potential treatment strategy. For this reason, future research on the impact of cachexia and sarcopenia in patients with NSCLC should also focus on the different treatment schedules now available. Further analyses including the comparison of survival of patients who have received chemotherapy versus immunotherapy with cachexia and/or sarcopenia are needed to better suit the current clinical landscape.

Recent studies have suggested that cachexia and sarcopenia increase toxicity in patients treated with chemotherapy [18,50]. Yet, in patients treated with either TKI or ICI, there were mixed results on the effect of cachexia and sarcopenia on toxicity [20,43,51]. Our results demonstrated that toxicity was similar in patients with or without cachexia and/or sarcopenia, indicating that a radical intent treatment strategy in patients with cachexia and/or sarcopenia at baseline could be deemed as a safe option.

Our study is to our knowledge the first to explore the impact of cachexia and sarcopenia on survival in patients with sOMD NSCLC including a large and well-defined cohort of patients on an intention-to-treat basis.

Nonetheless, our study has some limitations; for example, even if we used the EORTC consensus for the definition of OMD to make our population homogeneous, information regarding the tumor volume is lacking. The study of the variability of both primary tumor volume and metastases volume may represent a useful prognostic biomarker and needs to be evaluated in future studies. Although our patients were homogeneous for their oligometastatic state and intention-to-treat basis, they have different oncogenic drivers and they received different treatment schedules. These factors may impact the survival, representing a weakness of our results. Other limitations of our study are the retrospective nature and the lack of an external validation for the classification of cachexia and for the measurement of sarcopenia. In fact, a consensus regarding the optimal way to evaluate those factors still does not exist, making it difficult to compare results. Also, PMI was used as a surrogate for sarcopenia and sarcopenia was not based on SMI, and muscle strength and function (i.e., handgrip strength), which are validated by the EWGSOP [35]. Specific cut-offs and standardized methods to evaluate sarcopenia are still lacking; for this reason, we used the unbiased sex-specific lower 25th percentile as a cut-off, but a better standardization, such as an external validation, is mandatory in the future. Furthermore, patients in our cohort were in relatively good clinical condition, as most patients had a WHO-PS of 0–1 (89.9%). Therefore, we could not compare the detrimental effect of WHO-PS ≥2 in comparison with cachexia and sarcopenia as other studies have shown [23,25]. Additionally, response to induction systemic therapy, serum CRP level, and older age at diagnosis (≥75 years) could be used as a prognostic factor instead of cachexia or sarcopenia, and in turn they could play a fundamental role in selecting patients with sOMD fit for radical treatment. Finally, the different subgroups of patients classified according to the presence or absence of sarcopenia and cachexia are small. Therefore our results are still preliminary and larger studies are needed to confirm these hypothesis-generating data, even if our analysis on an intention-to-treat basis represents an important strength of our results despite the small subgroups.

## 5. Conclusions

Cachexia and sarcopenia, either separately or combined, were not associated with an inferior survival in patients with sOMD NSCLC, irrespective of receiving LRT. High CRP and older age at diagnosis (≥75 years) were associated with inferior survival and could be potential prognostic factors, helping the decision-making process of clinicians in selecting the patients who may benefit from the addition of LRT. Further research and prospective studies should evaluate highly needed potential biomarkers, to better identify those patients who are most likely to benefit from the addition of LRT at the diagnosis in a homogenous population.

## Figures and Tables

**Figure 1 cancers-16-00230-f001:**
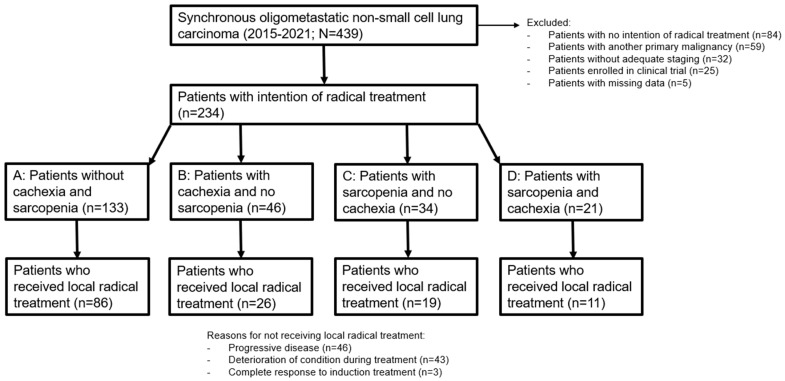
Flowchart of patient inclusion and selection.

**Figure 2 cancers-16-00230-f002:**
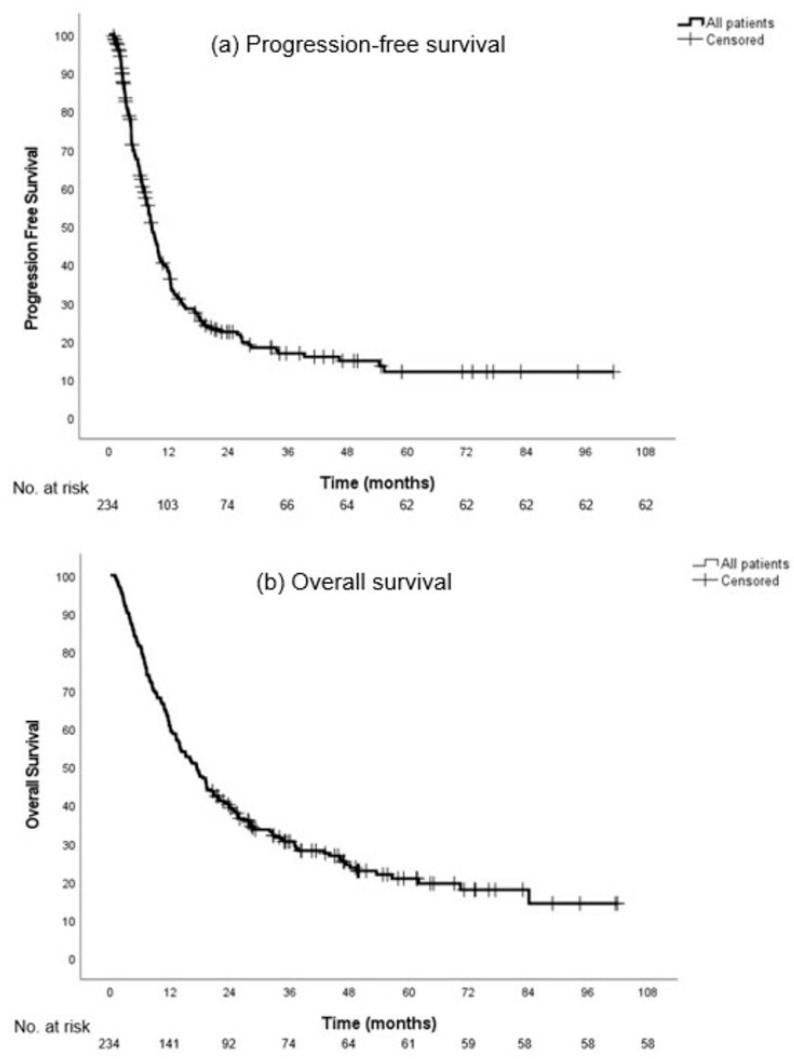
Progression-free survival (**a**) and overall survival (**b**) in all patients.

**Figure 3 cancers-16-00230-f003:**
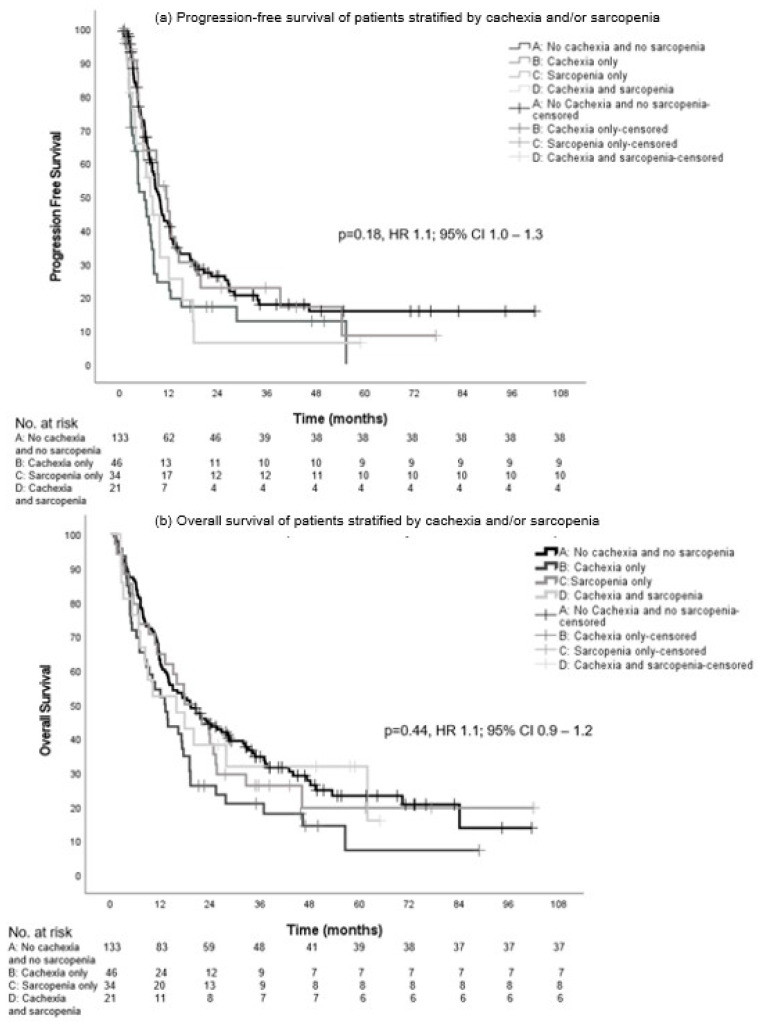
Progression-free survival (**a**) and overall survival (**b**) in patients with intention of LRT stratified by cachexia and/or sarcopenia.

**Figure 4 cancers-16-00230-f004:**
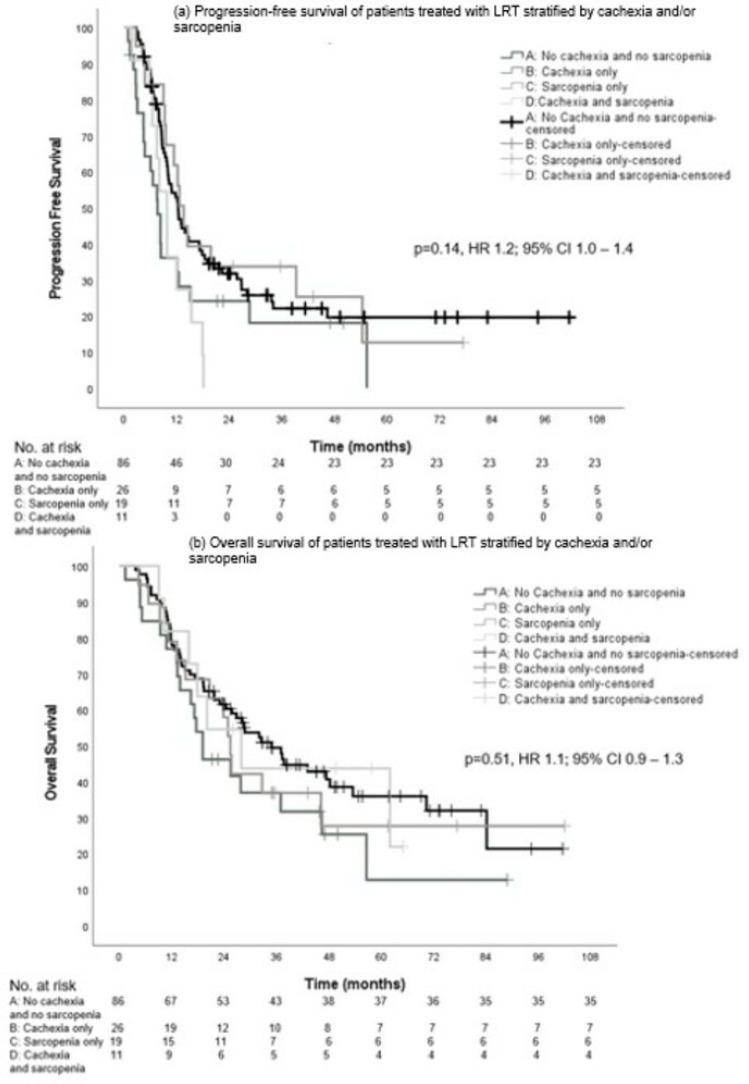
Progression-free survival (**a**) and overall survival (**b**) in patients who received LRT stratified by cachexia and sarcopenia.

**Figure 5 cancers-16-00230-f005:**
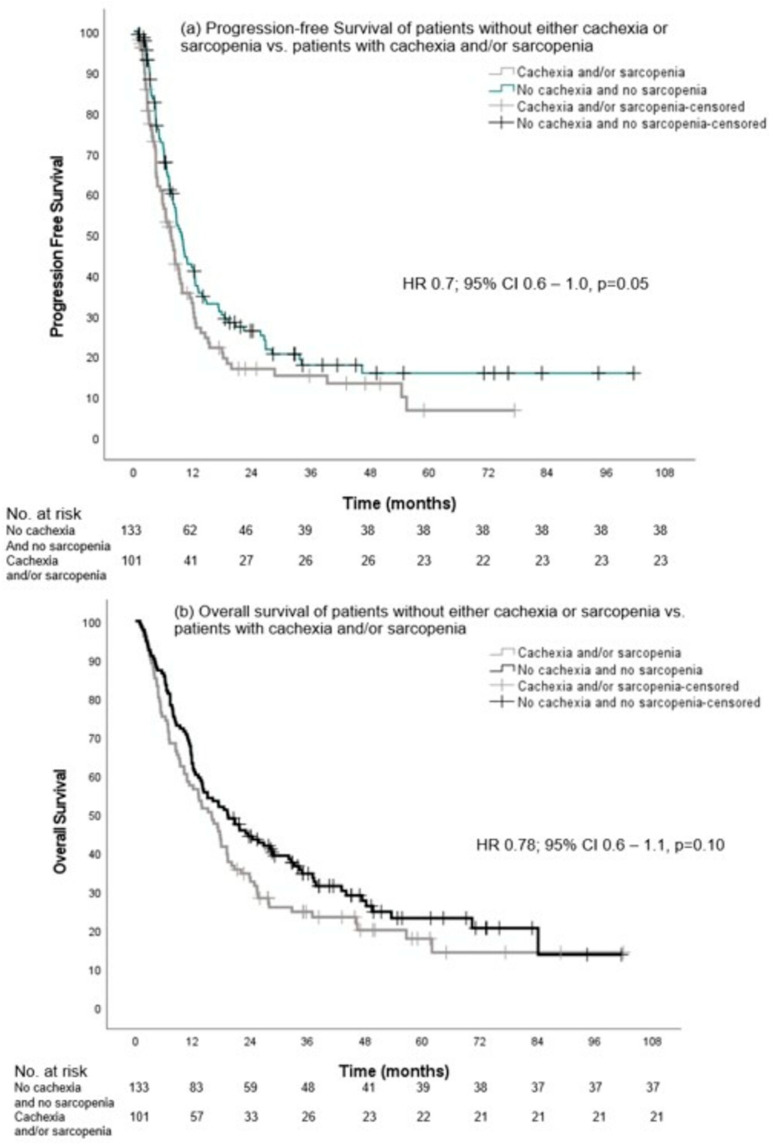
Progression-free survival and overall survival of patients without either cachexia or sarcopenia vs. patients with cachexia and/or sarcopenia.

**Table 1 cancers-16-00230-t001:** Effect of cachexia or sarcopenia on survival rates in different stages of NSCLC in recent studies.

Authors	Setting	Study Design	No. of Patients	Cachexia or Sarcopenia	Definition of Cachexia/Sarcopenia	Treatment	Outcome (Cachexia/Sarcopenia vs. Non-Cachexia/Sarcopenia)
Madeddu et al. [19]	Advanced NSCLC	Prospective, monocenter	74	Cachexia and sarcopenia	Cachexia: 1. Weight loss ≥ 5% during the past 6 months or weight loss of more than 2% and BMI < 20 2. MiniCASCO questionnaire [22] sarcopenia: SMI at L3, women < 39 cm^2^/m^2^; men < 55 cm^2^/m^2^	ICI treatment	Cachexia was an independent predictor of negative survival in patients treated with ICI. Sarcopenia was not predictive of the PFS and OS in patients treated with ICI.
Matsuo et al. [23]	Advanced or recurrent NSCLC	Retrospective, monocenter	183	Cachexia	Weight loss ≥ 5% during the past 6 months or weight loss of more than 2% and BMI < 20	PD-1/PD-L1 inhibitors	Significantly shorter median PFS (2.1 vs. 5.1 months, *p* < 0.001) and OS (5.6 vs. 15.0 months, *p* < 0.001). Multivariate analysis: cachexia in combination with poor PS is associated with worse survival.
Morimoto et al. [24]	All stages of NSCLC (81% stage III–IV), and recurrent	Retrospective, multicenter	196	Cachexia	Weight loss ≥ 5% during the past 6 months or weight loss of more than 2% and BMI < 20	Chemoimmunotherapy	Overall population: significantly shorter median PFS (6.7 vs. 9.3 months, *p* = 0.04), and no significant difference in OS. In PD-L1 ≥ 50% expression: no significant difference in PFS and OS.
Burtin et al. [25]	Stage I–III NSCLC	Prospective, monocenter	936	Sarcopenia	Low FFMI (women < 15 kg/m^2^; men < 17 kg/m^2^) [26] and handgrip weakness [27]	Primary RT, sequential CRT or concurrent CRT	Patients with PS 0–1: handgrip weakness and low FFMI were significant prognostic factors for OS. Patients with PS ≥ 2: handgrip weakness and low FFMI were not related to OS.
Bolte et al. [28]	Advanced NSCLC	Retrospective, monocenter	92	Sarcopenia	Sex-specific 25th percentile of the PMI (at L3) in the cohort	Chemoimmunotherapy	Significantly shorter median OS (9.1 vs. 22.3 months, *p* = 0.003).
Hasenauer et al. [29]	Stage I–III NSCLC	Retrospective, monocenter	401	Sarcopenia	SMI at L3. Sarcopenia: women < 38.5 cm^2^/m^2^; men < 52.4 cm^2^/m^2^	VATS pulmonary resection	Sarcopenia is associated with a higher rate of postoperative complications and longer hospital stay.
Karaman et al. [30]	Stage III NSCLC	Retrospective, monocenter	56	Sarcopenia	SMI at L3. Sarcopenia: women < 38.5 cm^2^/m^2^; men < 52.4 cm^2^/m^2^	Concurrent CRT or RT only	Significantly shorter median OS (19.0 vs. 38.0 months, *p* < 0.04).
Katsui et al. [31]	Stage III NSCLC	Retrospective, monocenter	60	Sarcopenia	SMI and PMI at L3, cut-off values based on time-dependent ROC curve	Concurrent CRT	Shorter 1,3,5-year OS rates in patient with low SMI (63.6%, 53.8% and 17.9%) vs. high SMI (92.1%, 59.6% and 51.0%).
Lyu et al. [20]	Advanced NSCLC	Retrospective, monocenter	131	Sarcopenia	SMI at L3. Sarcopenia: women < 31.6 cm^2^/m^2^; men < 40.2 cm^2^/m^2^	EGFR-TKI or ICI	Shorter median PFS (6.4 vs. 15.1 months, *p* < 0.001) and OS (13.0 vs. 26.0 months, *p* < 0.001). Sarcopenia was an independent predictor of poor OS and PFS.
Yuan et al. [32]	Stage III–IV NSCLC	Retrospective, monocenter	202	Sarcopenia	SMI at L3. Sarcopenia: women < 32.5 cm^2^/m^2^; men 44.7 cm^2^/m^2^	Treatment according NCCN guidelines	Significantly shorter median PFS (8.0 vs. 13.1 months, *p* = 0.02) and OS 13.3 months vs. 25.8 months, *p* = 0.003).

Abbreviations: NSCLC—non-small cell lung cancer; BMI—body mass index; CASCO—cachexia score; SMI—skeletal mass index; ICI—immune checkpoint inhibitor; PFS—progression-free survival; OS—overall survival; PD-L1—programmed death ligand 1; PS—performance status; FFMI—fat-free mass index; RT—radiation therapy; CRT: chemoradiation; PMI—Psoas muscle index; VATS—video-assisted thoracic surgery; ROC—receiver operating characteristic; EGFR—epidermal growth factor receptor; TKI—tyrosine kinase inhibitor; NCCN—National Comprehensive Cancer Network.

**Table 2 cancers-16-00230-t002:** Baseline clinical characteristics.

Characteristic	All Patients (*n* = 234)	A: No Cachexia and No Sarcopenia (*n* = 133)	B: Cachexia, No Sarcopenia (*n* = 46)	C: Sarcopenia, No Cachexia (*n* = 34)	D: Cachexia and Sarcopenia (*n* = 21)	*p* Value
Median age at diagnosis (range), years	67 (39–89)	68 (39–89)	67 (49–85)	65 (48–88)	66 (50–85)	0.69
Sex (%)						0.82
Male	123 (52.6)	68 (51.1)	25 (54.3)	20 (58.8)	10 (52.4)	
Female	111 (47.4)	65 (48.9)	21 (45.7)	14 (41.2)	11 (47.6)	
WHO-PS (%)						0.001
0	102 (43.6)	72 (54.1)	12 (26.1)	16 (47.1)	2 (9.5)	
1	108 (46.2)	51 (38.3)	26 (56.5)	15 (44.1)	16 (76.2)	
2	21 (9.0)	10 (7.5)	6 (13.0)	2 (5.9)	3 (14.3)	
3 *	3 (1.3)	0 (0)	2 (4.3)	1 (2.9)	0 (0)	
Median BMI (range), kg/m^2^	25.0 (15.8–42.1)	26.4 (18.4–42.1)	23.0 (16.6–35.9)	24.5 (16.5–41.4)	21.1 (15.8–31.2)	<0.001
Median weight loss in last 6 months before diagnosis (range), kg	0 (0–25)	0 (0–5)	7.3 (2–25)	0 (0–9)	9 (1–25)	<0.001
Median PMI (range), mm^2^/m^2^	6.32 (1.50–14.58)	7.05 (3.90–12.49)	6.67 (3.78–14.58)	4.21 (1.50–5.94)	3.95 (2.13–5.89)	<0.001
Median LDH	208 (120–699)	208 (120–699)	196 (132–499)	211 (122–406)	201 (151–415)	0.82
Median CRP mg/L	13 (1–268)	12 (1–174)	14 (1–268)	13 (1–164)	22 (1–217)	0.14
Median serum albumin (range), g/dL	39.0 (21.0–53.0)	39.7 (21.2–53.0)	37.8 (22.1–47.0)	39.0 (29.5–49.0)	36.0 (21.0–45.4)	0.005
Median serum total protein g/dL	70.4 (57.0–81.0)	69.1 (57.0–76.0)	73.7 (67.0–81.0)	67.0 (57.0–71.0)	76.1 (76.1–76.1)	0.26
Smoking status						0.02
Current	101 (43.2)	44 (33.1)	28 (60.9)	17 (50.0)	12 (57.1)	
Former	118 (50.4)	79 (59.4)	15 (32.6)	16 (47.1)	8 (38.1)	
Never	12 (5.1)	9 (6.7)	1 (2.2)	1 (2.9)	1 (4.8)	
Unknown	3 (1.3)	1 (0.8)	2 (4.3)	0 (0)	0 (0)	
NSCLC subtype (%)						0.83
Non-squamous	183 (78.2)	102 (76.6)	36 (78.3)	27 (79.4)	18 (85.7)	
Squamous	51 (21.8)	31 (23.3)	10 (21.7)	7 (20.6)	3 (14.3)	
PD-L1 status (%)						0.57
Positive (≥50%)	68 (29.1)	34 (25.6)	12 (26.1)	14 (41.2)	8 (38.1)	
Positive (1–49%)	45 (19.2)	28 (21.1)	6 (13.0)	7 (20.6)	4 (19.0)	
Negative (<1%)	39 (16.7)	24 (18.0)	8 (17.4)	3 (8.8)	4 (19.0)	
Unknown	82 (35.0)	47 (35.3)	20 (43.5)	10 (29.4)	5 (23.9)	
Driver mutation (%)						
ALK+	1 (0.4)	0 (0)	0 (0)	1 (2.9)	0 (0)	0.12
BRAF+	5 (2.1)	3 (2.3)	1 (2.2)	0 (0)	1 (4.8)	0.70
EGFR +	11 (4.7)	8 (6.0)	0 (0)	2 (5.9)	1 (4.8)	0.41
KRAS+	62 (26.5)	34 (25.6)	13 (28.3)	9 (26.5)	6 (28.6)	0.98
RET+	2 (0.9)	1 (0.8)	0 (0)	0 (0)	1 (4.8)	0.21
ROS1+	3 (1.3)	0 (0)	2 (4.3)	1 (2.9)	0 (0)	0.10
T stage ** (%)						0.24
x	10 (4.3)	9 (6.8)	0 (0)	1 (2.9)	0 (0)	
1a	5 (2.1)	2 (1.5)	1 (2.2)	2 (5.9)	0 (0)	
1b	14 (6.0)	8 (6.0)	4 (8.7)	1 (2.9)	1 (4.8)	
1c	21 (9.0)	13 (9.8)	3 (6.5)	5 (14.7)	0 (0)	
2a	28 (12.0)	17 (12.8)	3 (6.5)	7 (20.6)	1 (4.8)	
2b	17 (7.3)	10 (7.5)	3 (6.5)	1 (2.9)	3 (14.3)	
3	47 (20.1)	29 (21.8)	7 (15.2)	6 (17.6)	5 (23.8)	
4	92 (39.3)	45 (33.8)	25 (54.3)	11 (32.4)	11 (52.4)	
N stage (%)						0.35
0	70 (29.9)	44 (33.1)	9 (19.6)	13 (38.2)	4 (19.0)	
1	16 (6.8)	6 (4.5)	4 (8.7)	4 (11.8)	2 (9.5)	
2	85 (36.3)	51 (38.3)	16 (34.8)	10 (29.4)	8 (38.1)	
3	63 (26.9)	32 (24.1)	17 (37.0)	7 (20.6)	7 (33.3)	
Median number of metastases	1	1	1	1	2	0.32
Number of metastases (%)						
1	127 (54.3)	76 (57.1)	24 (52.2)	18 (52.9)	9 (42.9)	0.65
2	69 (29.5)	42 (31.6)	10 (21.7)	9 (26.5)	8 (38.1)	0.47
3	20 (8.5)	7 (5.3)	7 (5.3)	3 (8.8)	3 (14.3)	0.15
4	12 (5.1)	4 (3.0)	3 (6.5)	4 (11.8)	1 (4.8)	0.21
5	6 (2.6)	4 (3.0)	2 (4.3)	0 (0)	0 (0)	0.54
Metastatic sites (%)						
Brain	95 (40.6)	49 (36.8)	17 (37.0)	17 (50.0)	12 (57.1)	0.20
Bone	45 (19.2)	32 (24.1)	8 (17.4)	3 (8.8)	2 (9.5)	0.13
Adrenal	30 (12.8)	15 (11.3)	7 (15.2)	5 (14.7)	3 (14.3)	0.89
Lung	64 (27.4)	37 (27.8)	13 (28.3)	8 (23.5)	6 (28.6)	0.96
Nodal (extrathoracic)	32 (13.7)	18 (13.5)	9 (19.6)	3 (8.8)	2 (9.5)	0.51
Pleural	12 (5.1)	3 (2.3)	3 (6.5)	4 (11.8)	2 (9.5)	
Liver	8 (3.4)	4 (3.0)	3 (6.5)	1 (2.9)	0 (0)	0.54
Soft tissue	6 (2.6)	4 (3.0)	0 (0)	1 (2.9)	1 (4.8)	0.63
Renal	2 (0.9)	0 (0)	1 (2.2)	1 (2.9)	0 (0)	0.26
Subcutis	1 (0.4)	0 (0)	1 (0.6)	0.53	0 (0)	0.86
Peritoneal	1 (0.4)	0 (0)	1 (2.2)	0 (0)	0 (0)	0.25
Type of systemic treatment						
Chemotherapy	155 (79.5)	91 (81.2)	30 (83.3)	23 (79.3)	11 (61.1)	0.67
Chemotherapy + ICI	18 (9.2)	12 (10.7)	3 (8.3)	1 (3.5)	2 (11.1)	0.23
ICI monotherapy	12 (6.2)	4 (3.6)	1 (2.8)	3 (10.3)	4 (22.2)	0.01
TKI	10 (5.1)	5 (4.5)	2 (5.6)	2 (5.9)	1 (5.6)	0.96
Best response after induction systemic treatment according to RECIST 1.1 (%)						0.93
CR	3 (1.3)	2 (1.5)	1 (2.2)	0 (0)	0 (0)	
PR	103 (44.0)	65 (48.9)	16 (34.8)	12 (35.3)	10 (47.6)	
SD	57 (24.2)	29 (21.8)	12 (26.1)	12 (35.3)	4 (19.0)	
PD	46 (19.7)	23 (17.3)	12 (26.1)	6 (17.6)	5 (23.8)	
Unknown	25 (10.7)	14 (10.5)	5 (10.9)	4 (11.8)	6 (10.9)	
Actual radical treatment (%)	142 (60.7)	86 (64.7)	26 (56.5)	19 (55.9)	11 (52.4)	0.54

Abbreviations: WHO-PS—World Health Organization Performance Score; BMI—body mass index; PMI—Psoas muscle index; CRP—C-reactive protein; PD-L1—programmed death ligand 1; ALK—anaplastic lymphoma kinase; EGFR—epidermal growth factor receptor; ICI—immune checkpoint inhibitor; TKI—tyrosine kinase inhibitor; RECIST 1.1—response evaluation criteria in solid tumors 1.1; CR—complete response; PR—partial response; SD—stable disease; PD—progressive disease. * Three patients had a WHO-PS of 3 due to symptomatic cerebral edema, but their clinical condition rapidly improved after treatment and were deemed candidates for LRT by the MDT. ** TNM staging is based on TNM 8 (AJCC). If staging was based on TNM 7 in the MDT, it is recalculated into TNM 8.

**Table 3 cancers-16-00230-t003:** Logistic regression analysis for one-year overall survival for all patients.

Characteristics		Univariate Analysis		Multivariate Analysis	
		Hazard Ratio (95% CI)	*p*-Value	Hazard Ratio (95% CI)	*p*-Value
Age (ref: <75 years)	≥75 years	2.5 (1.3–4.9)	0.005	2.1 (1.1–4.3)	0.03
Gender (ref: male)	female	0.5 (0.3–0.9)	0.02	0.6 (0.3–1.0)	0.07
WHO-PS (ref: 0–1)	2–3	2.0 (0.8–4.6)	0.12	1.7 (0.7–4.2)	0.24
Smoking (ref: never)	Former	1.5 (0.5–5.0)	0.50		
	Current	1.7 (0.5–5.7)	0.37		
Histology (ref: non-squamous)	Squamous	0.4 (0.2–0.8)	0.01	0.6 (0.3–1.1)	0.09
BMI (ref: <25 kg/m^2^)	≥25 kg/m^2^	1.2 (0.7–2.0)	0.49		
Cachexia		0.7 (0.4–1.2)	0.17	0.6 (0.4–1.2)	0.15
Sarcopenia	≥6.05 mm^2^/m^2^ for men and ≥4.20 mm^2^/m^2^ for women	1.0 (0.5–1.8)	0.91	1.0 (0.5–1.8)	0.92
Serum albumin (ref: <40 g/L)	≥40 g/L	0.8 (0.5–1.5)	0.55		
Serum CRP (ref: ≤5 mg/L)	>5 mg/L	1.8 (0.9–3.5)	0.09		
Serum LDH (ref: <248)	≥248	1.2 (0.7–2.1)	0.59		
Actionable driver mutation * (ref: yes)		0.8 (0.3–2.1)	0.64		

Abbreviations: ref—reference; WHO-PS—World Health Organization Performance Score; BMI—body mass index; CRP—C-reactive protein; LDH—lactate dehydrogenase; *—actionable: ALK, BRAF V600, EGFR, ROS1, RET. KRAS (G12C) was not actionable at the time of patient inclusion.

**Table 4 cancers-16-00230-t004:** Logistic regression analysis for one-year progression-free survival for all patients.

Characteristics		Univariate Analysis		Multivariate Analysis	
		Hazard Ratio (95% CI)	*p*-Value	Hazard Ratio (95% CI)	*p*-Value
Age (ref: <75 years)	≥75 years	1.7 (0.8–3.6)	0.16	1.9 (0.8–5.1)	0.17
Gender (ref: male)	female	0.6 (0.3–0.9)	0.04	0.6 (0.3–1.2)	0.13
WHO-PS (ref: 0–1)	2–3	2.6 (0.8–7.8)	0.10	2.5 (0.5–12.0)	0.25
Smoking (ref: never)	Former	2.4 (0.5–11.4)	0.28		
	Current	2.6 (0.6–12.2)	0.25		
Histology (ref: non-squamous)	Squamous	1.0 (0.5–1.9)	0.91		
BMI (ref: <25 kg/m^2^)	≥25 kg/m^2^	1.3 (0.7–2.2)	0.43		
Cachexia		0.5 (0.2–1.0)	0.05	0.4 (0.2–1.0)	0.05
Sarcopenia	≥6.05 mm^2^/m^2^ for men and ≥4.20 mm^2^/m^2^ for women	0.9 (0.5–1.8)	0.84	1.3 (0.6–2.8)	0.58
Serum albumin (ref: <40 g/L)	≥40 g/L	0.83 (0.5–1.6)	0.56		
Serum CRP (ref: ≤5 mg/L)	>5 mg/L	2.8 (1.4–5.4)	0.003	3.6 (1.3–5.2)	0.008
Serum LDH (ref: <248)	≥248	1.0 (0.5–1.8)	0.94		
Actionable driver mutation (ref: yes) *		0.6 (0.2–1.9)	0.36		

Abbreviations: ref—reference; WHO-PS—World Health Organization Performance Score; BMI—body mass index; CRP—C-reactive protein; LDH—lactate dehydrogenase; *—actionable: ALK, BRAF V600, EGFR, ROS1, RET. KRAS (G12C) was not actionable at the time of patient inclusion.

**Table 5 cancers-16-00230-t005:** Logistic regression analysis for one-year overall survival for patients who received LRT.

Characteristics		Univariate Analysis		Multivariate Analysis	
		Hazard Ratio (95% CI)	*p*-Value	Hazard Ratio (95% CI)	*p*-Value
Age (ref: <75 years)	≥75 years	1.5 (0.5–4.2)	0.45	0.8 (0.2–3.4)	0.87
Gender (ref: male)	female	0.5 (0.3–1.2)	0.09	1.7 (0.6–4.5)	0.29
WHO-PS (ref: 0–1)	2	1.1 (0.2–5.4)	0.93	0.8 (0.1–5.0)	0.82
Smoking (ref: never)	Former	0.8 (0.1–7.9)	0.86		
	Current	1.0 (0.1–10.0)	0.98		
Histology (ref: non-squamous)	Squamous	0.3 (0.1–0.8)	0.02	2.4 (0.8–7.0)	0.11
BMI (ref: <25 kg/m^2^)	≥25 kg/m^2^	2.1 (0.9–4.7)	0.09		
Cachexia		1.0 (0.4–2.4)	0.93	1.3 (0.4–3.7)	0.67
Sarcopenia	≥6.05 mm^2^/m^2^ for men and ≥4.20 mm^2^/m^2^ for women	1.1 (0.4–2.9)	0.87	0.7 (0.2–2.5)	0.55
Serum albumin (ref: <40 g/L)	≥40 g/L	1.7 (0.7–4.1)	0.29		
Serum CRP (ref: ≤5 mg/L)	>5 mg/L	1.9 (0.6–5.6)	0.25	0.6 (0.2–1.9)	0.40
Serum LDH (ref: <248)	≥248	1.3 (0.5–3.1)	0.58		
TRAE (ref: toxicity ≤ 2)	Toxicity grade ≥ 3	1.5 (0.7–3.4)	0.32		
Best response to induction systemic therapy (ref: CR and PR)	SD and PD	2.2 (0.9–5.1)	0.06		
Actionable driver mutation (ref: yes) *		0.89 (0.3–3.5)	0.87		

Abbreviations: ref—reference; WHO-PS—World Health Organization Performance Score; BMI—body mass index; CRP—C-reactive protein; LDH—lactate dehydrogenase; TRAE—treatment related adverse events; CR—complete response; PR—partial response; SD—stable disease; PD—progressive disease; *—actionable: ALK, BRAF V600, EGFR, ROS1, RET. KRAS (G12C) was not actionable at the time of patient inclusion.

**Table 6 cancers-16-00230-t006:** Logistic regression analysis for one-year progression-free survival for patients who received LRT.

Characteristics		Univariate Analysis		Multivariate Analysis	
		Hazard Ratio (95% CI)	*p*-Value	Hazard Ratio (95% CI)	*p*-Value
Age (ref: <75 years)	≥75 years	0.9 (0.4–2.2)	0.77	1.1 (0.3–3.9)	0.84
Gender (ref: male)	female	0.7 (0.4–1.4)	0.35	1.2 (0.5–2.7)	0.75
WHO-PS (ref: 0–1)	2	3.4 (0.7–16.7)	0.14	0.6 (0.1–3.0)	0.50
Smoking (ref: never)	Former	1.4 (0.2–9.0)	0.71		
	Current	1.2 (0.2–7.9)	0.82		
Histology (ref: non-squamous)	Squamous	1.1 (0.5–2.4)	0.90		
BMI (ref: <25 kg/m^2^)	≥25 kg/m^2^	1.0 (0.9–1.1)	0.82		
Cachexia		0.5 (0.2–1.1)	0.09	2.7 (1.0–6.9)	0.05
Sarcopenia	≥6.05 mm^2^/m^2^ for men and ≥4.20 mm^2^/m^2^ for women	1.2 (0.5–2.6)	0.73	0.4 (0.2–1.3)	0.13
Serum albumin (ref: <40 g/L)	≥40 g/L	1.2 (0.6–2.5)	0.63		
Serum CRP (ref: ≤5 mg/L)	>5 mg/L	2.8 (1.2–6.7)	**0.02**	0.4 (0.1–0.9)	0.02
Serum LDH (ref: <248)	≥248	1.2 (0.6–2.5)	0.61		
TRAE (ref: toxicity ≤ 2)	Toxicity grade ≥ 3	1.0 (0.5–2.0)	0.99		
Best response to induction systemic therapy (ref: CR and PR)	SD and PD	1.5 (0.7–3.1)	0.26		
Actionable driver mutation (ref: yes) *		2.2 (0.3–18.6)	0.46		

Abbreviations: ref—reference; WHO-PS—World Health Organization Performance Score; BMI—body mass index; CRP—C-reactive protein; LDH—lactate dehydrogenase; TRAE—treatment related adverse events; CR—complete response; PR—partial response; SD—stable disease; PD—Progressive disease; *—actionable: ALK, BRAF V600, EGFR, ROS1, RET. KRAS (G12C) was not actionable at the time of patient inclusion.

## Data Availability

The data presented in this study are available upon request from the corresponding author.

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
