# Peer review of "Cachexia and Sarcopenia in Oligometastatic Non-Small Cell Lung Cancer: Making a Potential Curable Disease Incurable?"

_cancers, 2024, doi:10.3390/cancers16010230_

Round 1
Reviewer 1 Report
Comments and Suggestions for Authors
The article "Cachexia and sarcopenia in oligometastatic non-small cell lung cancer: Making a potential curable disease incurable? by Bartolomeo describes the studies on cachexia and sarcopenia in metastatic non-small cell lung cancer. They indicate that these functional markers are not significant for NSCLC with minimal metastasis which they have termed as oligometastatic cancer. Other biomarkers identified as statistically significant such as WHO-PS, median BMI, median weight loss etc. are known in the literature. Thus, information on the two functional biomarkers in the subcategory of NSCLC is valuable.
This is a retrospective study of subjects presented in the radiology/oncology board of their hospital from 2015 to 2021. While they could identify 439 potential subjects, they could assess only 234 subjects for this study. Of these 133 did not have cachexia or sarcopenia.
Subjects with late-stage NSCLC generally have cachexia and sarcopenia which are markers of poor prognosis. Here the authors want to determine whether these two biologically functional markers have utility as biomarkers in oligometastasis. The numbers for individual categories: cachexia alone (46), sarcopenia alone (34) and cachexia and sarcopenia together (21) are small in comparison to non-cachexia non sarcopenia subjects (133) to derive a statistical conclusion.
The figure in supplementary data includes PFS and OS for non-cachexia and non-sarcopenia (133) vs cachexia and/or sarcopenia (101). Here PFS is statistically significant. The two figures in the supplementary need to be included as main figures. This information of statistical significance for PFS need to be included in the simple summary, abstract and results.
The oligometastases is not well defined. The median number of another organ cancer seems to be 1 which is mostly brain, lung, and bone. What is not clear is the size of these tumors. Are there number of micro metastases within these organs or a single large metastatic tumor? This information could provide information on variations within this category of metastasis and their importance as significant biomarkers.
Author Response
Dear editor,
Thank you for reviewing our manuscript entitled “Cachexia and sarcopenia in oligometastatic non-small cell lung cancer: Making a potential curable disease incurable?”
We thank the reviewers for their comments, questions and suggestions. We think that our manuscript has improved and we hope that our revised article will be suitable for publication in Cancers. We have responded to each of the issues raised and you can find below our point-by-point response to the reviewers’ comments.
We look forward to hearing from you.
Kindest regards,
On behalf of all the authors,
Dirk K.M. De Ruysscher.
Reviewer 1:
The article "Cachexia and sarcopenia in oligometastatic non-small cell lung cancer: Making a potential curable disease incurable? by Bartolomeo describes the studies on cachexia and sarcopenia in metastatic non-small cell lung cancer. They indicate that these functional markers are not significant for NSCLC with minimal metastasis which they have termed as oligometastatic cancer. Other biomarkers identified as statistically significant such as WHO-PS, median BMI, median weight loss etc. are known in the literature. Thus, information on the two functional biomarkers in the subcategory of NSCLC is valuable.
Answer: we thank the reviewer for recognizing that our work is valuable.
This is a retrospective study of subjects presented in the radiology/oncology board of their hospital from 2015 to 2021. While they could identify 439 potential subjects, they could assess only 234 subjects for this study. Of these 133 did not have cachexia or sarcopenia.
Subjects with late-stage NSCLC generally have cachexia and sarcopenia which are markers of poor prognosis. Here the authors want to determine whether these two biologically functional markers have utility as biomarkers in oligometastasis. The numbers for individual categories: cachexia alone (46), sarcopenia alone (34) and cachexia and sarcopenia together (21) are small in comparison to non-cachexia non sarcopenia subjects (133) to derive a statistical conclusion.
Answer: we agree with the reviewer that the numbers are small, therefore we stressed in the discussion section that our results are hypothesis generating.
The figure in supplementary data includes PFS and OS for non-cachexia and non-sarcopenia (133) vs cachexia and/or sarcopenia (101). Here PFS is statistically significant. The two figures in the supplementary need to be included as main figures. This information of statistical significance for PFS need to be included in the simple summary, abstract and results.
Answer: We agree regarding the figures in the supplemental material and we added them in the main paper (Figure 5). This information was already mentioned in the results section but now it is also added in the abstract and in the simple summary.
The oligometastases is not well defined. The median number of another organ cancer seems to be 1 which is mostly brain, lung, and bone. What is not clear is the size of these tumors. Are there number of micro metastases within these organs or a single large metastatic tumor? This information could provide information on variations within this category of metastasis and their importance as significant biomarkers.
Answer: we do not agree that we did use a not well defined definition of oligometastases as we used the EORTC definition. To avoid selection bias, we screened all patients diagnosed with NSCLC in the years of inclusion, and we selected all patients fulfilling the EORTC definition of synchronous oligometastatic disease. Therefore, we think that the median number of metastases and their localization reflects the natural behavior of oligometastatic NSCLC, with no selection bias.
Unfortunately, we do not have the information regarding the tumor volume and this may represent a weakness of our study, for this reason we added the following to the discussion: “Nonetheless, our study has some limitations, for example, even if we used the EORTC consensus for the definition of OMD to make our population homogeneous, information regarding the tumor volume is lacking. The study of the variability of both primary tumor volume and metastases volume may represent a useful prognostic biomarker and needs to be evaluated in future studies.”
Reviewer 2 Report
Comments and Suggestions for Authors
The authors reported a retrospective analysis (n=234) to investigate the association of cachexia and/or sarcopenia with survival in patients with adequately staged synchronous oligometastatic disease. They obtained clinical data from the medical records of two Dutch hospitals from 2015 to 2021. The study was approved by the ethics committee (2021-2973 and Z2021168). The hypothesis was unique and reasonable. However, I have substantial concerns about inadequate sample size, high heterogeneity, and inappropriate statistical methodology. Accordingly, I do not support publication of this manuscript.
Detailed Comments:
-
This cohort has a large oncological heterogeneity (e.g., driver mutations, PD-L1 status, and type of chemotherapy). Each of these factors strongly influences OS, PFS, and the decision of local radical treatment. However, the authors did not evaluate these associations or include them in the multivariate model.
-
The subgroups of interest (cachexia/sarcopenia) have sample sizes that are too small to test the authors' hypothesis. It may be better to focus only on the presence/absence of cachexia.
-
The multivariate analysis should only include pre-treatment factors because the authors hypothesized to use them for pre-treatment decision-making. It is wrong to include post-treatment factors such as best response or adverse events in the multivariate models.
-
Their assessment of sarcopenia was not standardized using the arbitrary cutoff.
-
The external validation is needed to propose a new hypothesis.
Author Response
Dear editor,
Thank you for reviewing our manuscript entitled “Cachexia and sarcopenia in oligometastatic non-small cell lung cancer: Making a potential curable disease incurable?”
We thank the reviewers for their comments, questions and suggestions. We think that our manuscript has improved and we hope that our revised article will be suitable for publication in Cancers. We have responded to each of the issues raised and you can find below our point-by-point response to the reviewers’ comments.
We look forward to hearing from you.
Kindest regards,
On behalf of all the authors,
Dirk K.M. De Ruysscher.
Reviewer 2:
The authors reported a retrospective analysis (n=234) to investigate the association of cachexia and/or sarcopenia with survival in patients with adequately staged synchronous oligometastatic disease. They obtained clinical data from the medical records of two Dutch hospitals from 2015 to 2021. The study was approved by the ethics committee (2021-2973 and Z2021168). The hypothesis was unique and reasonable. However, I have substantial concerns about inadequate sample size, high heterogeneity, and inappropriate statistical methodology. Accordingly, I do not support publication of this manuscript.
Answer: we thank the reviewer for recognizing the unique dataset. We hope we have sufficiently addressed the concerns below.
Detailed Comments:
- This cohort has a large oncological heterogeneity (e.g., driver mutations, PD-L1 status, and type of chemotherapy). Each of these factors strongly influences OS, PFS, and the decision of local radical treatment. However, the authors did not evaluate these associations or include them in the multivariate model.
Answer: All the patients of the current study were selected according to the consensus from the European Organisation for Research and Treatment of Cancer (EORTC), considering as oligometastatic a maximum of five metastases in a maximum of three different organs and making our size very homogeneous within the field of oligometastatic disease. Moreover, all the patients received adequate and homogeneous staging and our study was conducted on an intention-to-treat basis. Therefore, our population is very homogeneous in the field of oligometastatic disease. However, we do agree with your comment that it is not homogenous for the oncogenic drivers and type of treatment. For this reason, we added the following to our discussion: “Although our patients were homogeneous for their oligometastatic state and intention-to-treat basis, they have different oncogenic drivers and they received different treatment schedules. These factors may impact on the survival, representing a weakness of our results”
- The subgroups of interest (cachexia/sarcopenia) have sample sizes that are too small to test the authors' hypothesis. It may be better to focus only on the presence/absence of cachexia.
Answer: we agree with the small subgroups and that this is hypothesis generating. This has been added to the discussion section. However, in the originally submitted manuscript we already also performed a comparison between patients with no cachexia and no sarcopenia versus patients with cachexia and/or sarcopenia (now the figure was added in the manuscript as Figure 5), with a trend to longer PFS in the first group. Moreover, we think that the intention-to-treat basis may represent an important strength of our analysis despite the small subgroups. We tried to clearly mention these considerations adding the following to the discussion: “Finally, the different subgroups of patients classified according to the presence or absence of sarcopenia and cachexia are small and larger studies are needed to confirm these hypothesis generating data, even if our analysis on an intention-to-treat basis represents an important strength of our results despite the small subgroups”.
Finally, as showed in the main flowchart, during the screening several patients were excluded for different reasons, such as inadequate staging or no intention of local radical treatment.
- The multivariate analysis should only include pre-treatment factors because the authors hypothesized to use them for pre-treatment decision-making. It is wrong to include post-treatment factors such as best response or adverse events in the multivariate models.
Answer: We agree with your suggestion and we removed treatment related adverse effects and better response to induction systemic treatment in the logistic regression that we have performed so far (such as in the abstract and in the discussion). However, we think that these two factors may be useful as prognostic factors in the decision strategy to select those patients who are most likely to benefit from the addition of local radical treatment. For this reason, we performed a new multivariate analysis for 1-year progression free survival and overall survival selecting only patients who received LRT. The new multivariate analyses are described in the manuscript and reported in the Tables 5 and 6.
- Their assessment of sarcopenia was not standardized using the arbitrary cutoff.
Answer: Regarding the assessment of sarcopenia, available studies during these years used different definitions of cachexia and sarcopenia, as shown in the table 1. However, for our study we evaluated sarcopenia according to the the European Working Group on Sarcopenia in Older People (EWGSOP) definition, measuring the musculature through the psoas muscle at level of L3. Then we calculated the PMI by dividing the psoas muscle cross-sectional area divided by the height, as stated by Gomez-Perez et all. We agree that validated cut offs are still lacking in literature and for this reason, as stated in material and methods, we used the sex specific lower 25th percentile but for sure specific cutoff and external validation are needed. In the previous version of the manuscript, we have already stated that: “Other limitations of our study are the retrospective nature and the lack of an external validation for the classification of cachexia and for the measurement of sarcopenia. In fact, a consensus for the optimal way to evaluate those factors still does not exist, making it difficult to compare results. Also, sarcopenia was only defined using PMI and not based on SMI, and muscle strength and function (i.e handgrip strength), which are validated by the EWGSOP [35]” . However, to emphasize this important limitation, in the new version we added the following: “Specific cut off and standardized methods to evaluate sarcopenia are still lacking, for this reason we used the unbiased sex specific lower 25th percentile as cut off but a better standardization, such as an external validation, is mandatory in the future
- The external validation is needed to propose a new hypothesis.
Answer: we agree that more data is needed, therefore we already added to our discussion that are results are hypothesis generating.
Reviewer 3 Report
Comments and Suggestions for Authors
The results are interesting and the trial was thoroughly performed
however 80% of the patients did not recieve immunotherapy which is probably explained by the date of the trial, Accumulated data ( see for example a recent review
(2023) Sarcopenia as a Determinant of the Efficacy of Immune Checkpoint Inhibitors in Non-Small Cell Lung Cancer: A Meta-Analysis, Nutrition and Cancer, 75:2, 685-695, DOI: 10.1080/01635581.2022.2153879
does suggest that immunotherapy efficacy is negatively effected by sarcopenia thus i would expect a more thorough discussion of the current research result in comparison with the previous studies described above
is it a matter of utilizing different therapies? is it because of the different population of patients which do not include ecog 2 population? etc
this is extremely important if one wants to utilize the presented results for current clinical practice where most patients without an actionable gene mutation are treated with immunotherapy or chemoimmuno
still i was impressed with all the results and the effort put in the trial and thus i would suggest publishing this article following much more thorough discussion of it s limitation in the current clinical set up
Author Response
Dear editor,
Thank you for reviewing our manuscript entitled “Cachexia and sarcopenia in oligometastatic non-small cell lung cancer: Making a potential curable disease incurable?”
We thank the reviewers for their comments, questions and suggestions. We think that our manuscript has improved and we hope that our revised article will be suitable for publication in Cancers. We have responded to each of the issues raised and you can find below our point-by-point response to the reviewers’ comments.
We look forward to hearing from you.
Kindest regards,
On behalf of all the authors,
Dirk K.M. De Ruysscher
Reviewer 3:
The results are interesting and the trial was thoroughly performed
Answer: thank you for these compliments
however 80% of the patients did not recieve immunotherapy which is probably explained by the date of the trial, Accumulated data ( see for example a recent review (2023) Sarcopenia as a Determinant of the Efficacy of Immune Checkpoint Inhibitors in Non-Small Cell Lung Cancer: A Meta-Analysis, Nutrition and Cancer, 75:2, 685-695, DOI: 10.1080/01635581.2022.2153879 does suggest that immunotherapy efficacy is negatively effected by sarcopenia thus i would expect a more thorough discussion of the current research result in comparison with the previous studies described above
is it a matter of utilizing different therapies? is it because of the different population of patients which do not include ecog 2 population? etc
this is extremely important if one wants to utilize the presented results for current clinical practice where most patients without an actionable gene mutation are treated with immunotherapy or chemoimmuno
still i was impressed with all the results and the effort put in the trial and thus i would suggest publishing this article following much more thorough discussion of it s limitation in the current clinical set up
Answer: We agree that this is an important factor to take into account in our analysis and in the impact of our results on the clinical practice. However, it is unknown whether the impact of cachexia and /or sarcopenia is the same for oligometastatic patients with a good performance status versus patients with more widespread disease and a potentially poorer performance status. For this reason, after the discussion regarding ICI and TKI, we added the following: “However, as we collected patients treated between 2015 to 2021 (in part before the widespread introduction of ICIs), in our population only 15% of patients received treatment with ICIs and 10% of patients received treatment with TKI. This may represent a relevant limitation to interpret our data in the current clinical scenario, as it has been suggested that the efficacy of ICI is less in patients with widespread metastatic NSCLC treated with ICI. For patients with oligometastatic disease and a good performance status, data is lacking. In fact, for patients who are candidate to receive ICIs, immunotherapy represents a safe and with long term tumor control potential treatment strategy. For this reason, future research on the impact of cachexia and sarcopenia in patients with NSCLC should also focus on the different treatment schedules now available. Further analyses including the comparison of survival of patients who have received chemotherapy vs immunotherapy with cachexia and/or sarcopenia are needed to better suit the current clinical landscape”.
Round 2
Reviewer 1 Report
Comments and Suggestions for Authors
Authors have addressed the concerns raised and have included limitations of their study due to small number of individual categories. A statement that it is a preliminary report could be included in the abstract, and discussion .
Author Response
Reviewer 1:
Authors have addressed the concerns raised and have included limitations of their study due to small number of individual categories. A statement that it is a preliminary report could be included in the abstract, and discussion.
Answer: Thank you for appreciating our changes. Following your suggestion, in this new round of review we clearly stated the preliminary nature of our results both in the abstract and in the discussion.
Reviewer 2 Report
Comments and Suggestions for Authors
Dear authors,
Thank you for the revised manuscript. I think that many of the comments have been addressed and adequately reflected in the revised manuscript. However, we need the following revisions.
Thank you very much.
-
The impact of driver mutations on PFS and OS should never be ignored. Therefore, the presence or absence of the driver mutation should be added to the multivariate variables.
-
What the authors call "sarcopenia" differs from the EWGSOP definition of sarcopenia. Sarcopenia/non-sarcopenia should be replaced by low/high PMI.
Author Response
Reviewer 2:
Dear authors,
Thank you for the revised manuscript. I think that many of the comments have been addressed and adequately reflected in the revised manuscript.
Answer: Thank you for your comment, we are happy that you appreciated our new work and the way we improved the manuscript following your previous revisions.
However, we need the following revisions.
Thank you very much.
- The impact of driver mutations on PFS and OS should never be ignored. Therefore, the presence or absence of the driver mutation should be added to the multivariate variables.
Answer: We agree with your consideration and we added driver mutation in the univariate analysis with no significant results. For this reason, we did not consider driver mutations in the multivariate analysis. You can find the new analysis in the results.
- What the authors call "sarcopenia" differs from the EWGSOP definition of sarcopenia. Sarcopenia/non-sarcopenia should be replaced by low/high PMI
Answer: We understand the concerns regarding the definition of sarcopenia, as also previous studies used different definitions and cut-off. For this reason, in this new version we clearly stated in the simple summary, in the introduction, material and methods and in the discussion that we used the PMI as surrogate of sarcopenia to overcome this problem and to avoid confusion.
Reviewer 3 Report
Comments and Suggestions for Authors
i think they authors have addressed in their added paragraph my main issue with the article
Author Response
Reviewer 3:
I think they authors have addressed in their added paragraph my main issue with the article
Answer: Thank you very much. Your suggestions during the first round of review were appreciated and improved our paper.
Round 3
Reviewer 2 Report
Comments and Suggestions for Authors
Dear Authors,
Thank you for your revised manuscript. All the comments are responded to and adequately reflected in the revised manuscript. This manuscript has markedly improved. Therefore, I endorse the publication of this excellent manuscript. Thank you for all your efforts.
Best regards,